# Oxygen evolution reaction dynamics monitored by an individual nanosheet-based electronic circuit

Peiyao Wang[1,2], Mengyu Yan[1,3], Jiashen Meng[1], Gengping Jiang[2,4], Longbing Qu[2], Xuelei Pan[1], Jefferson Zhe Liu [2] & Liqiang Mai [1]

The oxygen evolution reaction involves complex interplay among electrolyte, solid catalyst, and gas-phase and liquid-phase reactants and products. Monitoring catalysis interfaces between catalyst and electrolyte can provide valuable insights into catalytic ability. But it is a challenging task due to the additive solid supports in traditional measurement. Here we design a nanodevice platform and combine on-chip electrochemical impedance spectroscopy measurement, temporary *I-V* measurement of an individual nanosheet, and molecular dynamic calculations to provide a direct way for nanoscale catalytic diagnosis. By removing $O_2$ in electrolyte, a dramatic decrease in Tafel slope of over 20% and early onset potential of 1.344 V vs. reversible hydrogen electrode are achieved. Our studies reveal that $O_2$ reduces hydroxyl ion density at catalyst interface, resulting in poor kinetics and negative catalytic performance. The obtained in-depth understanding could provide valuable clues for catalysis system design. Our method could also be useful to analyze other catalytic processes.

[1] State Key Laboratory of Advanced Technology for Materials Synthesis and Processing, Wuhan University of Technology, Wuhan 430070, China. [2] Department of Mechanical and Aerospace Engineering, Monash University, Clayton, Victoria 3800, Australia. [3] Department of Material Science and Engineering, University of Washington, Seattle, Washington 98195-2120, USA. [4] College of Science, Wuhan University of Science and Technology, Wuhan 430081, China. Correspondence and requests for materials should be addressed to M.Y. (email: ymymiles@whut.edu.cn) or to J.Z.L. (email: zhe.liu@monash.edu) or to L.M. (email: mlq518@whut.edu.cn)

Water electrolysis has been regarded as an environmentally friendly route to hydrogen gas production[1–3]. However, efficiency of water catalysis is severely limited by poor kinetics of the oxygen oxidation reaction, namely oxygen evolution reaction (OER)[3, 4]. OER is a complex process that involves the interplay among solid catalysts, electrolyte, gas-phase and liquid-phase reactants, and products[5, 6]. During an OER process, chemical reactions mainly take place at catalysts/electrolyte interface. Unfortunately, there are insufficient in-depth understanding of the reaction interfaces[7]. The difficulty mainly arises from two experimental limitations: first, the interfaces for catalytic reaction are generally buried between solid and liquid phases, which are very difficult to access and detect by conventional spectroscopic techniques;[8] second, active species are mixed with binders and conductive carbon additives, which hinder an accurate investigation of the electrochemical interfaces in traditional measurements[8, 9].

Previously, most of fundamental studies focused on rate-determining steps and identification of catalytic activity descriptor for the OER from thermodynamic aspect[10–12]. There have been significant advances on relationships between material's catalytic activities and its electronic structure, aided by a wealth of spectroscopic techniques and first-principles computations[13, 14]. However, there are much fewer studies on OER kinetic process, particularly those processes taking place in an electrode/electrolyte interface region. The distribution of ions and water at the interface determine the kinetics of mass and electron transfer process. Our understanding is still quite limited. Oxygen is a product of OER. In terms of reaction equilibrium (i.e., Le Chatelier's principle), it is known that $O_2$ concentration increase in electrolyte (with the ongoing of OER) hinders the catalytic reaction. But there is no clear answer whether oxygen molecules would affect OER kinetics. The initial OER catalytic steps involve hydroxide formation at active surface sites through the discharge of a hydroxide ion ($* + OH^- \rightarrow *OH + e^-$) in alkaline solution[7]. It is reasonable to expect that oxygen molecules could adsorb at the reaction interfaces, hindering the formation of hydroxide and hence the catalytic kinetic performance. It is fundamentally important to investigate this hypothesis and gain in-depth understanding.

Currently, a single nanostructure electrochemical device has been exploited as a powerful tool to investigate intrinsic electrochemical processes and properties at nanoscale[15–20]. Several unique advantages enable such a nanodevice to be a great platform to directly probe the OER processes for reliable information. First, a single nanowire/nanosheet can be used as the working electrode, avoiding influences of the binders and conductive carbon additives[20–22]. Second, structure and composition of the nanodevice can be precisely designed and controlled, which is essential to eliminate experimental uncertainties and allows quantitatively or semi-quantitatively fundamental studies[19]. Third, it is much easier to monitor the change of physical properties of individual nanowire and nanosheet (such as conductivity) in a nanodevice. This advantage is largely unexplored up to date. It could allow us to probe interface properties during the OER process directly.

In this work, we design a single electrochemical nanodevice consisting of nickel nanoparticle catalysts anchored on a graphene nanosheet electrode. We propose and use concurrent measurement of electrical conductivity of the electrode materials to probe the effects of $O_2$ at the reaction interface during the OER. The on-chip electrochemical impedance spectroscopy (EIS) measurement and molecular dynamics (MD) simulations are also carried out. Our results show that oxygen in electrolyte has an inhibition effect on OER performance. A significant decrease in Tafel slope over 20% and an early onset potential of 1.344 V (with reference to RHE) are observed by removing $O_2$ in electrolyte. Our study reveals that oxygen adsorption at catalytic interface would reduce $OH^-$ ion concentration in the double layer (DL) and thus result in a poor kinetics and negative catalytic performance.

## Results

**Device fabrication.** Figure 1c depicts the Ni-graphene-based electrochemical device. The fabrication procedure is schematically

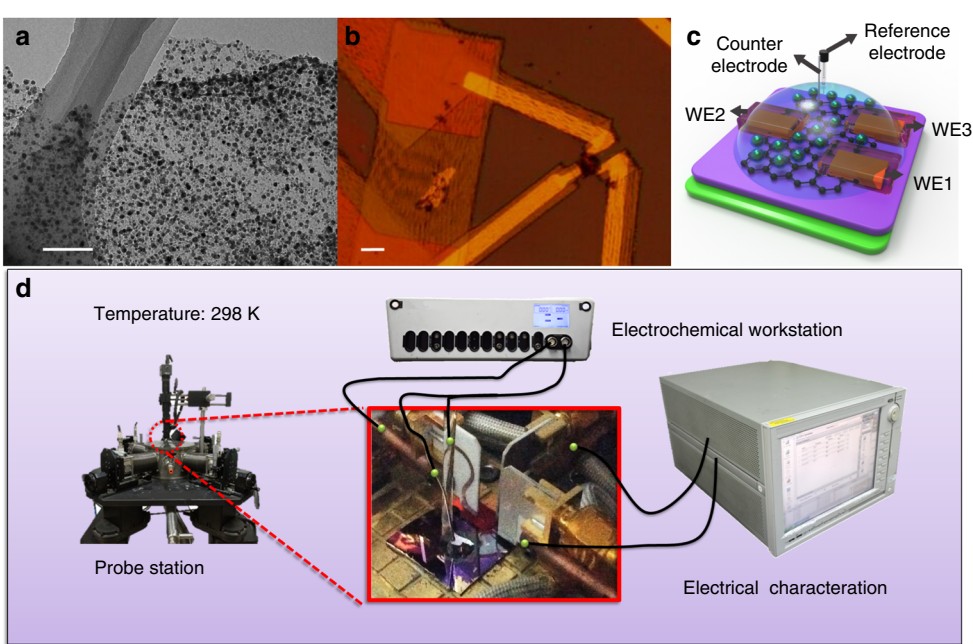

**Fig. 1** Working principle of temporal electrical transport measurement. **a** TEM image of as-prepared Ni-graphene nanosheets with a *scale bar* at 200 nm. **b** Optical microscope image of nanosheets contacted with three metallic electrodes with a *scale bar* at 10 μm. **c** Schematic illustration of the Ni-graphene nanosheet-based device with a microscopic electrochemical cell on it. **d** Illustration of the device (*inset*) and corresponding measurement equipment layout with the three-dimensional view

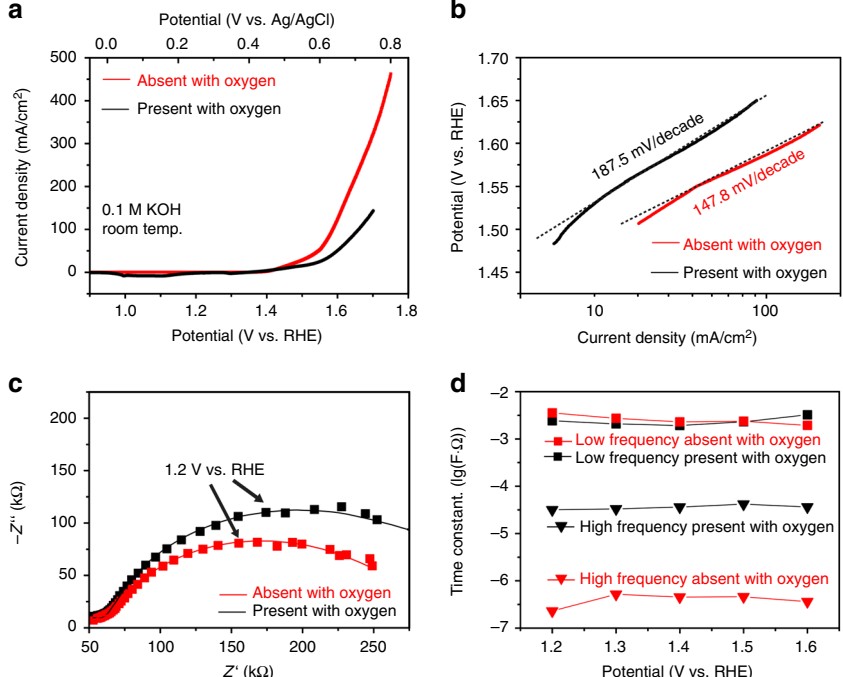

**Fig. 2** OER activity of Ni-graphene nanosheet-based device. **a** Oxygen evolution currents of Ni-graphene nanosheet measured in 0.1 M KOH. **b** Tafel plots of OER currents in **a**. **c** The Nyquist plots in the potential at 1.2 V vs. RHE together with the responding fitted curve based on the equivalent circuit model. **d** Plot of the high- and low-frequency time constants

illustrated in Supplementary Figs. 1 and 2 and Supplementary Note 1. The Ni-Graphene nanosheets were prepared through a hydrothermal method and subsequent heat treatment (Supplementary Note 2). Then, the Ni-graphene nanosheets were selectively deposited onto Si/SiO$_2$ wafer. Three metallic electrodes (WE1, WE2, and WE3) were deposited on the device using electron beam lithography followed by Au/Cr (150 nm/5 nm) evaporation and a lift-off process (Fig. 1b). An insulating photoresist (SU8-2002) layer was used to cover the metallic electrodes to prevent leakage current to the aqueous electrolyte. The insulating property of the photoresist and SiO$_2$ was verified through electrical resistance measurement. We found that the resistance of photoresist layer as well as the SiO$_2$ (Supplementary Fig. 3) was around four orders of magnitudes larger than that of Ni-graphene nanosheet, indicating that leakage currents passing through SiO$_2$ and photoresist should be negligible and would not interfere the electric and electrochemical signals during later electrochemical measurements. In addition, to check whether there was parasitic reaction in our nanodevice, a comparable device in which there were no active catalysts while other parts were kept the same as the original one was fabricated and tested (Supplementary Fig. 4). No reaction peaks were observed.

**Electrochemical performance**. A number of Ni particles with diameter around 22.6 nm (Supplementary Note 3 and Supplementary Fig. 5) were anchored on graphene surface (Fig. 1a and Supplementary Fig. 6). Figure 1d shows the experimental setup. Figure 2a shows that the onset potential is 1.380 V vs. RHE in electrolyte with saturated O$_2$. By removing oxygen in electrolyte (see Methods section), the onset potential is reduced to 1.344 V vs. RHE and a high current density of 10 mA cm$^{-2}$ was measured at a low potential of 1.438 V vs. RHE. To check whether the results obtained from this nanodevice is consistent with conventional powder sample performance, Supplementary Fig. 7a–c compare OER performance of Ni catalysts that were prepared in conventional ways. There were clear performance

gaps between oxygen-absence and oxygen-presence conditions. They qualitatively agree with the results from our nanodevice. We also found that the nanosheet had a higher current density than that of traditional powder samples at a given potential. This could be attributed to a large effective surface area and enhanced electron transport (shorter distance between active sites and metallic electrodes)[9, 23–26]. For the non-iR-corrected Tafel plots of the polarization curves (Fig. 2b), the Ni-graphene nanosheet under oxygen-absence condition shows a smaller Tafel slope (147.8 mV/decade) than that under oxygen-presence condition (187.5 mV/decade), showing a faster kinetic process. It indicates the critical effect of O$_2$ on the reaction kinetics.

To gain more understanding, on-chip EIS measurement was carried out. Figure 2c shows the results at an electrochemical potential of 1.2 V vs. RHE. A two constant parallel model[27] was adopted to analyze the EIS data (Supplementary Figs. 8–10). At low frequency, the time constants ($\tau$) were almost the same under both conditions. However, two orders of magnitude decrease of $\tau$ at high frequency is observed for the oxygen-absence case (Fig. 2d, Supplementary Note 4). This should be contributed to the reduced charge transfer resistance at interface[27–30]. But EIS data could not provide insights to understand such a reduction.

**Resistance testing and analysis**. We designed and performed on-chip concurrent $I$–$V$ measurement of the Ni-graphene electrode to directly probe the interface property. A series of temporal $I$–$V$ measurements were carried out during both oxidation (from 0 to 0.7 V vs. Ag/AgCl) and subsequent reduction (from 0.7 to 0 V vs. Ag/AgCl) process[31]. The evolution of the conductivity (Fig. 3a) and electrode structure (Supplementary Fig. 11) is reversible, indicating a reliably repeatable performance of this nanodevice. Figure 3b, c and Supplementary Fig. 12 show that the resistance from $I$–$V$ results is significantly different under oxygen-absence and oxygen-presence conditions, which correlates with the OER performance gaps (Fig. 2). With the

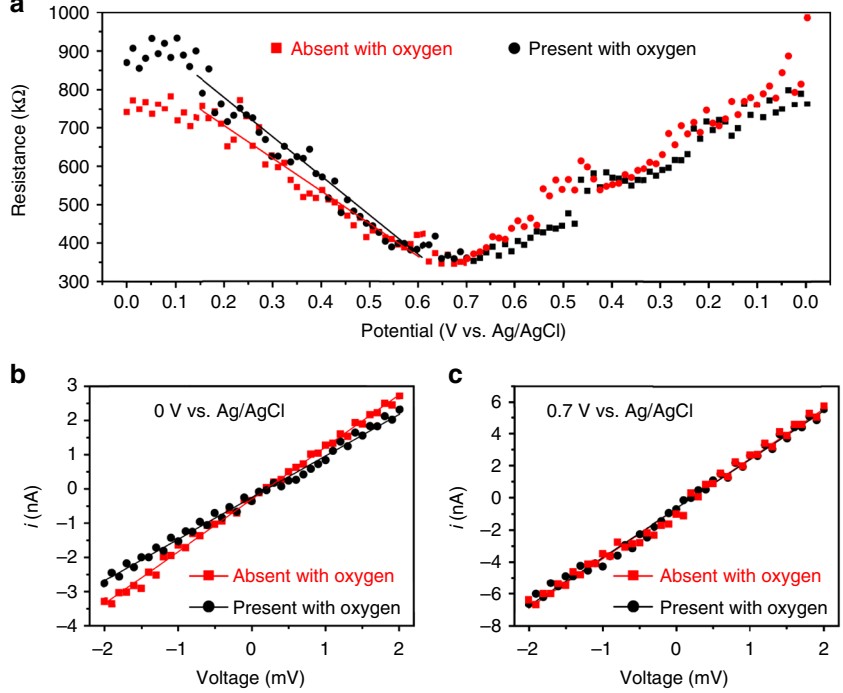

**Fig. 3** Temporal electrical transport measurement of the Ni-graphene nanosheet-based device. **a** Resistance (under potential bias 2 mV) vs. electrochemical potentials trace corresponding to Fig. 2a. **b** I–V characteristics of a typical Ni-graphene nanosheet-based device at potential of 0 V vs. Ag/AgCl. **c** I–V characteristics of a typical Ni-graphene nanosheet-based device at potential of 0.7 V vs. Ag/AgCl

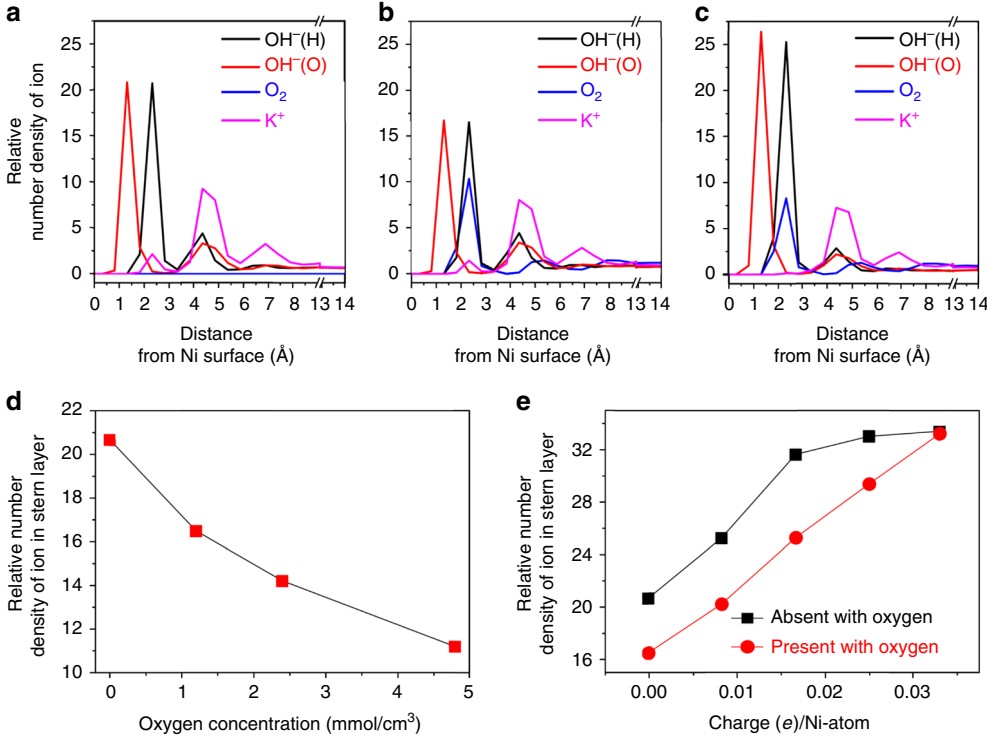

**Fig. 4** MD simulation results. **a** Relative number density $\rho$ of different electrolyte ions as a function of distance from the Ni cathode surface with oxygen concentration of 0. **b** Relative number density $\rho$ of different electrolyte ions as a function of distance from the Ni cathode surface with oxygen concentration of 0.12 mmol/cm$^3$. **c** Relative number density $\rho$ of different electrolyte ions as a function of distance from the Ni cathode surface with oxygen concentration of 0.12 mmol/cm$^3$ and with a charge of +0.0083e/Ni-atom. **d** The relative number density $\rho$ of OH$^-$ ions as a function of the concentration of oxygen in the electrolyte. **e** The relative density of OH$^-$ ions in the stern layer as a function of the charge number for per Ni atom

presence of oxygen in electrolyte, the initial resistance is 17% higher than that in the electrolyte without $O_2$.

Our I–V measurement approach has several advantages over EIS. First, our EIS data show nearly no differences at the highest frequency under oxygen-presence and oxygen-absence conditions. It could not provide sufficient resolution to make a comparison of the electrode resistance under these two conditions. Second, EIS measurement is time-consuming, which is not feasible for in situ measurement. Third, a relative complicated equivalent circuit model is required to interpret EIS results.

The working principle behind our I–V measurement is the gating effect of graphene. The electrochemical potential applied on our device can be decomposed into two parts (Supplementary Fig. 13). One is applied on the electric double layer (EDL) at the electrode–electrolyte interface. The second one is the voltage applied on graphene. According to the electrochemical-gating model[31, 32], we have.

$$V_G = \frac{h \cdot v_F \cdot \sqrt{\pi n}}{e} + \frac{ne}{C_{dl}} \quad (1)$$

where $V_G$ is the electrochemical gate voltage vs. reference, $h$ is the reduced Plank's constant, $v_F$ is the Fermi velocity of the Dirac electron in nanosheets, $e$ is the electron charge, $n$ is the carrier density, and $C_{dl}$ is the double layer capacitance. The first term on the right of Eq. (1) describes the voltage applied on graphene and its relationship to carrier density $n$. The second term describes the relationship between EDL voltage and EDL capacitance[32].

We hypothesize that oxygen would lead to a reduced double layer capacitance ($C_{dl}$). As a result, the EDL voltage (second term in Eq. (1)) will increase. Provided a constant value of $V_G$ in our experiments, the carrier density $n$ in the electrode would drop (first term in Eq. (1)) and the electrode resistance increases. This could explain the observed electrode resistance differences between oxygen-presence and oxygen-absence conditions in our I–V experiments.

**MD simulation**. In order to understand how $O_2$ could influence $C_{dl}$, MD simulations were employed (Supplementary Note 5, Supplementary Fig. 14, Supplementary Tables 1 and 2). In our simulations, the electrolyte was 1 M KOH with or without oxygen molecules. The catalyst surface was Ni(111) plane. The electrolyte was maintained at 298 K using the Berendsen thermostat. MD simulation was run for 1.5 ns to ensure thermo-equilibrium of the system. The results generated during the last 1 ns were used to analyze the EDL structure near the Ni surface. Here, we split the electrolyte region into a set of bins (of 0.5 Å in width) along the direction (x) perpendicular to the Ni surface. The relative number densities $\rho(x)$ of OH⁻, K⁺, and $O_2$ were calculated as the ratio of volumetric number density in each bin over that in electrolyte bulk[33].

Figure 4a and b show the results on neutral Ni surface at an oxygen concentration of 0.0 and 1.2 mmol/cm³, respectively. The sharp peaks represent accumulation of the electrolyte ions and oxygen molecules next to the electrode surfaces. Compared with Fig. 4a, Fig. 4b clearly demonstrates that $O_2$ is adsorbed at interface, where the position is similar to the position of adsorbed OH⁻, and significantly hindered accumulation of OH⁻ in EDL. The increase of $O_2$ concentration leads to a monotonic decrease of OH⁻ density at the reaction interface (Fig. 4d, Supplementary Fig. 15), leading to the decrease of $C_{dl}$.

Considering Ni surface would be oxidized during the OER process, other MD simulations were performed to examine whether the surface oxidation would affect our observations based on the above pure Ni theoretical model. As it is difficult to determine oxidation degree at the Ni surface, we choose the fully oxidized case (NiO-Ni). The MD model and the unit cell structure of NiO-Ni are presented in Supplementary Fig. 16. The ions distribution next to the NiO-Ni surface under oxygen-presence condition is presented in Supplementary Fig. 17. Again, the oxygen molecules exhibit a strong adsorption near the surface, leading to a reduction of OH⁻ density (Supplementary Fig. 18). The detrimental effect of oxygen should stem from the strong van der Waals (vdW) force between O($O_2$) and Ni/NiO, which is nearly the same to the interaction between O(OH⁻) and Ni/NiO (Supplementary Table 3). The similar interaction force drives the $O_2$ and OH⁻ occupy similar positions next to the Ni/NiO surface. This reduces the amount of OH⁻ in the EDL.

During the OER process, the Ni surface should have positive charges. The Coulomb force could drive OH⁻ closer to Ni surface, eliminating the detrimental role of $O_2$. To examine this effect, a set of surface charges was induced in the system (Supplementary Fig. 19). Figure 4c shows the results of a charged Ni surface with +0.0083$e$/Ni-atom and oxygen concentration of 1.2 mmol/cm³. The Coulomb interaction clearly leads to an enhanced relative density of OH⁻ (Fig. 4c) in comparison with that in Fig. 4a. But the hindrance effect of oxygen still exists (Fig. 4e). In Fig. 4e, we summarize the OH⁻ relative density in the stern layer as a function of surface charge at oxygen concentration of 0 or 1.2 mmol/cm³. The difference is narrowing under a higher surface charge. This trend is consistent with the narrowing conductivity gap in the temporal I–V experimental results (Fig. 3c).

## Discussion

The previous studies conclude that the reaction steps in alkaline electrolyte for the OER process are as follows (the * represents active site on the metal surface)[34]:

$$4OH^- \leftrightarrow OH^* + 3OH^- + e^- \quad (2)$$

$$\leftrightarrow O^* + H_2O + 2OH^- + 2e^- \quad (3)$$

$$\leftrightarrow HOO^* + H_2O + OH^- + 3e^- \quad (4)$$

$$\rightarrow O_2 + 2H_2O + 4e^- \quad (5)$$

where Ni is not able to immediately oxidize water. Ni transforms into a highly reactive species in the alkaline solution and then generates $O_2$. From thermodynamic aspect, the increased $O_2$ concentration in reaction system leads to a higher Gibbs free energy, thereby increasing the onset potential under oxygen-presence condition. It could also be understood in terms of Le Chatelier's principle. However, the reaction kinetic processes are not considered in above two thermodynamic aspects. The kinetic process includes the transport of the reactants to an active site, the adsorption of the reactions to the active site, and the reaction of reactants to form an adsorbed product. Using EIS, temporal I–V measurement results, and MD simulation, we can conclude that $O_2$ can adsorb near the surface of catalyst driven by the strong vdW interaction. Their position overlaps with that of adsorbed OH⁻ ions, reducing the concentration of OH⁻ in EDL, thus slowing down the charge transfer process and OER reaction kinetics. Different from the Le Chatelier's principle, which provides a clue to understand reaction direction from thermodynamic perspective, our new understanding offers insights on how surface adsorbed $O_2$ could influence the reaction kinetics. It should be noted that our conclusion is similar to that in catalytic $N_2O$ gas decomposition, in which reaction rate could be improved for around three times by using an oxygen-selective membrane[35].

In conclusion, we developed a nano-electrochemical Ni-graphene device to investigate the effects of oxygen in the electrolyte on OER kinetics at catalyst interfaces. By removing $O_2$ in electrolyte, we observed a significant decrease in Tafel slope of over 20% and an early onset potential of 1.344 V vs. RHE. A temporal in-device electronic characteristic measurement was employed together with an on-chip EIS measurement and MD simulations. We conclude that the oxygen acts as a barrier to significantly reduce the concentration of $OH^-$ ions at catalyst surface, slowing down the charge transfer process and OER kinetics. This new insight could provide valuable clues to design high-performance catalyst systems. This work also presents a powerful nano-electrochemical device platform and a temporal $I–V$ measurement method to investigate the OER kinetic interface properties.

## Methods

**OER activity characterization and in-device EIS measurement**. The device was loaded on a probe station (Lake shore, PPTX). The OER activities were characterized with a three-electrode electrochemical system using an electrochemical workstation (Autolab PGSTAT 302N). The WE1, platinum (Pt), and Ag/AgCl/saturated KCl were used as the working, counter, and reference electrodes, respectively (Supplementary Note 6). Two different electrolytes were adopted with different contents of $O_2$, namely 0.1 M KOH under presence with $O_2$ or absence with $O_2$. Specifically, the electrolyte absent with $O_2$ was prepared using degassing method where $N_2$ was inserted into the 5 mL 0.1 M KOH electrolyte for 1 h to expel the $O_2$. This device was maintained at 298 K by using a model 336 cryogenic temperature controller. The cyclic voltammetry (CV) experiments at different scan rates (5–30 mV/s) and linear-sweep voltammograms (LSV) at a scan rate of 5 mV/s were performed. The catalyst was cycled until stable CV curves were obtained. After that the catalyst performance was measured in LSV testing[36]. The in-device EIS was controlled at different electrochemical working potentials and to characterize the effects of the oxygen on the catalytic interface. To ensure a complete characterization, the EIS measurements were recorded over four frequency decades, from 10 KHz to 1 Hz, with potential amplitude of 10 mV.

**Temporal $I–V$ measurement**. As outlined in Fig. 1d, a semiconductor parameter analyzer (Agilent, B1500A) was connected to the WE2 and WE3 electrode. The temporal $I–V$ measurement detected the electrical transport properties of Ni-graphene nanosheet during the electrolysis process under a tiny bias voltage (up to 2 mV). Supplementary Fig. 20 shows the schematic illustration of the concurrent measurement of this work, and the equivalent circuit model of the electrical transport spectroscopy (gate) measurement[37]. Note that all the samples were measured at an excitation current <20 nA and thus any heating effects should be negligible.

**Data availability**. The data that support the finding of this study are available from the corresponding authors on reasonable request.

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

## Acknowledgements

This work was supported by the National Basic Research Program of China (2013CB934103, 2012CB933003), the National Natural Science Foundation of China (51302203, 11525211, 51272197, 51502227), the International Science & Technology Cooperation Program of China (2013DFA50840), the Hubei Science Fund for Distinguished Young Scholars (2014CFA035), the National Science Fund for Distinguished Young Scholars and the Fundamental Research Funds for the Central Universities (2013-ZD-7, 2014-YB-02). This research/project was undertaken with the assistance of resources and services from the National Computational Infrastructure (NCI), which is supported by the Australian Government.

## Author contributions

L.M., M.Y. and P.W. designed the experiments. M.Y., P.W., J.M. and X.P. performed the experiments. J.Z.L., G.J. and P.W. performed the MD simulations. M.Y., P.W., J.Z.L., G.J. and L.Q. discussed the interpretation of results and co-wrote the paper. All authors discussed the results and commented on the manuscript.

## Additional information

**Competing interests:** The authors declare no competing financial interests.

