## [Peer Review File · Nature Communications]

Reviewers' comments:

Reviewer #1 (Remarks to the Author):

The paper reports an investigation of the OER on Ni catalyst supported on a graphene nanosheet electrode. The use of graphene nanosheet allows the authors to probe the OER without the use of binders of carbon and to monitor the electrode conductivity.

Even though the technique is innovative and opens up exciting opportunities, its application to the study of the role of O₂ on the OER and the interpretation of the results have several limitations as described below. For this reason, I do not believe the paper to be suitable for publication on Nature Communication.

As the device is new, the authors should provide the reader with more evidence that the obtained results are consistent with the ones obtained with a conventional setup. In particular, the detrimental role of O₂ should be compared with the results obtained for conventionally prepared Ni catalysts (with or without binder and carbon), in order to show that their catalyst is representative. Otherwise, we cannot exclude the effect to be specific to the setup used. Indeed, the higher current density reported compared to previous studies could either be due to the enhanced electron transport as speculated by the authors or to parasitic reactions in their setup. In contact with the electrolyte and at OER potential the Ni surface is likely oxidized and transformed to Ni oxide or oxyhydroxide. The question then arises on the relevance of the reported theoretical model, based on pure Ni. It is also not clear which is the driving force for O₂ adsorption at the surface under these conditions, which is the key point of the paper.

Also assuming that the model is realistic, it is not clear what the rationale of varying the substrate charge is. Which is the point that the authors want to make? The authors should also check the reported charges, as in the text and in the caption they are different in sign.

The English should be improved prior to publication.

Reviewer #2 (Remarks to the Author):

This is a very interesting paper of very high scientific quality, topicality and general interest. The measurement of oxygen evolution at a single nano sheet is very nice and the experimental data are of very high quality. This in itself adds to the quality of the paper.

The authors have noticed that the Tafel slope is reduced significantly if oxygen is excluded from the solution. They suggest that adsorbed oxygen substitutes for adsorbed hydroxyl ion and thus reduced the catalytic activity. This may well be so. However, It has been commonly observed in linear potential sweep measurements performed in aqueous solution in the active oxygen evolution region, that the Tafel slope increases as the over potential is increased. This increase has been attributed to gas bubble shielding effects which add a resistance to the interface thereby changing the Tafel slope. Indeed one can increase the potential window over which a low Tafel slope is observed by devising a route to dislodge and expel gas bubbles from the electrode surface. Consequently, I am not clear as to how the authors excluded the oxygen. Was it by simple degassing? The authors should elaborate here. Furthermore if the catalysis occurs primarily at edge sites rather than basal plane sites the reaction rate should depend on flake size. Has this effect been noted?

I am happy that the paper get accepted if these queries are addressed.

Reviewer #3 (Remarks to the Author):

The authors in this work propose to use on-chip electrochemical, EIS and I-V measurement to monitor the catalytic interface of Ni nanoparticle decorated graphene during OER reactions. The authors utilize a single sheet nanodevice with three working electrodes and necessary electrochemical gating setup to achieve the measurements. As a result, on-chip OER reactions, corresponding impedance and conductance measurements can be carried out in two different

electrolytes, with or without dissolved oxygen. Characterization of electrochemical interface during practically relevant electro catalytic reactions such as HER, OER, and ORR are indeed challenging, especially for in situ and in operando studies. On-chip based concurrent electrochemical and electrical characterization could be an effective strategy. Therefore, the topic of this work is of great novelty and significance, and fits the broad interests of Nat Comm readers.

Here are some of my concerns, comments, and suggestions, which should be addressed before this manuscript can be considered acceptable for publication:

1) Although it has been demonstrated previously in literature that measurements based on nanodevice can be helpful for the studies of electrocatalytic systems, the authors in this manuscript, however, did not offer a convincing argument that the experimental design and technique used in this work is better than approaches otherwise available. For example, the OER and EIS measurements can be easily done with the same material on a bulk electrode, which probably will offer the same conclusion revealing the effect of O₂ concentration on the catalyst performance and electrochemical interface. The I-V measurement is only available based on the on-chip platform. However the authors only declared that the result is in consistent with EIS. To clearly demonstrate the scientific significance of their approach, the authors are suggested to offer discussions on what different information the in-operando I-V or I-V_g measurements can reveal compared with EIS measurement, and on the analysis and interpretation of the data.

2) The significance of the major conclusion of this manuscript is not clearly discussed. The conclusion that O₂ level can affect the adsorption of OH and hence the overall performance of the OER catalysts is in accordance with the Le Chatelier's principle that as the end product, lower concentration of O₂ will indeed facilitate the OH to O₂ transition.

3) The electrical measurement should be precise and accurate even at the nA level. But the electrical data in Fig. 3 is very noisy even when compared with the similar works previously published by the same author (on the topic of single nanowire battery).

4) For the temporary (in situ) I-V measurement, could there be some interference of the current signals, since WE1 is used to carry out OER reaction (electrochemical current), and WE2 and WE3 is used to carry out conductive current, and the three electrodes are electrically connected by the nanosheet. If we view the system as an electrochemically gated transistor, then the OER current can be viewed as gate leakage current. Such current, concurrently measured with I-V current between WE2 and WE3, should be demonstrated. Unless the gate current through WE1 is several orders smaller than the conductive current through WE2 and WE3, then one need to take into account the current interference. Background subtraction may be necessary to reveal the true conductive current level.

This is also the reason why in situ electrical measurement is technically challenging, because when the reaction is in place, the concurrent electrical measurement is easily being interfered. Note that in Manish Chhowalla's work (Nat Mater, 2015, 15, 1003), similar experimental set-up was used to study the MoS₂ HER reactions, but all the I-V measurement that determines the contact resistance is done ex situ, considering the possible interference as well. This paper should also be an appropriate citation in line 119 of this manuscript as well. For another example, in Xiangfeng Duan's work (ACS Nano, 2016, 10, 9919), the interference of similar level electrochemical gating current on the concurrent conductive current is considered, and an equivalent circuit is used to help subtract the background.

5) In Fig. 3b and c, it looks like the I-V for O₂ and no O₂ are different at 0V while overlapping at 0.7V. Does this indicate that the difference of OH adsorption is different under no OER reactions (0V), while OH adsorption is the same when OER reaction occurs (0.7V)? By the way, the signs of voltage axis seem to be wrong.

6) Based on Equation (1), the physical model of this device is a graphene based transistor. Which

physical parameter in equation (1) is directly linked to the measured current change, and how? Based on the description in the manuscript, the authors attributed the current change to the change of carrier densities as well as change in double layer capacitance. Which one is more dominant? And if both are in the case, can these signals be conclusively linked to the event of OH adsorption on Ni nanoparticles? I feel that more detailed discussions on the origin of the signals, the interpretation of the data, and how the data is linked to the conclusions need to be provided.

7) The overall writing of the manuscript can be improved. Phrase like "...are rarely in-depth understood", "in traditional, ..." and some other sentences can be further polished for an improved quality. There are also minor errors in supporting materials that interfere with smooth reading and understanding of the materials. A few examples include: in Figure S1e, "Left off" should be "lift off"?; in caption of Figure S4, "rietveld" may be a typo; in figure s6- it is better to note which color represents which scan rate in Figure S6a and Figure S6b; in Figure S13 caption, "with oxygen in the electrolyte (left)" should be "without oxygen in the electrolyte (left)".

Response Letter

Response to Reviewer-1

We thank the reviewer for the thoughtful review and critical comments about our manuscript. We welcome the opportunity to address and clarify the issues raised in the reviewer's report, and believe that the additional data and revisions carried out to address the reviewer's comments substantially strengthen our revised manuscript. Our responses to the points raised in the report are as follows:

General Comment. The paper reports an investigation of the OER on Ni catalyst supported on a graphene nanosheet electrode. The use of graphene nanosheet allows the authors to probe the OER without the use of binders of carbon and to monitor the electrode conductivity. Even though the technique is innovative and opens up exciting opportunities, its application to the study of the role of O₂ on the OER and the interpretation of the results have several limitations as described below. For this reason, I do not believe the paper to be suitable for publication on Nature Communications.

Response to the General Comment. We thank the reviewer for recognizing novelty and potentials of the technique developed by us. In the revised version, new comprehensive experimental and simulation data are included (describe in detail below) to address all concerns of the reviewer. We thank the reviewer again for challenging us to improve this work.

Comment-1. As the device is new, the authors should provide the reader with more evidence that the obtained results are consistent with the ones obtained with a conventional setup. In particular, the detrimental role of O₂ should be compared with the results obtained for conventionally prepared Ni catalysts (with or without binder and carbon), in order to show that their catalyst is representative. Otherwise, we cannot exclude the effect to be specific to the setup used. Indeed, the higher current density reported compared to previous studies could either be due to the enhanced electron transport as speculated by the authors or to parasitic reactions in their setup.

Response to Comment-1. We thank the reviewer for this critical comment. We carried out conventional OER experiments using traditionally prepared Ni catalysts and compared with our nano-device. The possible 'specific effects' are excluded. In addition, we did experiments using a comparable nano-device to exclude the parasitic reactions in our new setup.

The results are shown below:

New Supplementary Figure 7. OER activity of bulk catalysts under conventional measurement. a. Oxygen evolution currents of Ni catalyst measured in 1M KOH. **b.** Oxygen evolution currents of Ni-graphene catalyst measured in 1M KOH. **c.** Oxygen evolution currents of NiO-graphene catalyst measured in 1M KOH.

In page 4, we added: “Figure S7 a-c compared OER performance of Ni catalysts that were prepared in conventional ways. There were clear performance gaps between oxygen-absence and oxygen-presence conditions. They qualitatively agree with the results from our nano-device.”

In page 8 of SI, we added “We also performed OER experiments using conventional setups under oxygen-presence and oxygen-absence conditions at a scan rate of 5 mV/s. For Ni catalyst (Fig. S7a), a current density of 10 mA/cm² was achieved at a potential of 0.71 V vs. SCE under oxygen-absence condition, which is lower than that under the oxygen-presence condition, *i.e.*, 0.742 V vs. SCE. In the case of Ni-graphene (Fig. S7b), a current density of 10 mA/cm² occurs at 0.593 V vs. SCE under oxygen-absence condition, whereas a higher potential of 0.627 V vs. SCE can be seen under oxygen-presence condition. A similar trend is also observed in the NiO-graphene catalysts (Fig. S7c). The results are consistent to our new setup.”

In addition, to exclude the effect of the parasitic reactions in the nano-device, another new experiment was carried out: a comparable nano-device sample was tested in which there was no active catalyst material but the other parts were kept the same with original nano-device. The cyclic voltammetry (CV) curve at a scan rate of 10 mV/s is shown below:

New Supplementary Figure 4. CV curve of a comparable nano-device sample.

There are no obvious reaction peaks in the data and the current density is only at around - 0.02-0.06 nA/cm², 7-8 orders of magnitudes smaller than tested OER current density (in mA/cm²). This demonstrates no distinctive parasitic reaction in our setup. We have clarified this point in revised manuscript and included new data in Supplementary Figure S4.

In page 4, we added “In addition, to check whether there was parasitic reaction in our nano-device, a comparable device in which there were no active catalysts while other parts were kept the same as the original one was fabricated and tested (Fig.S4). No reaction peaks were observed.”

Comment-2a. In contact with the electrolyte and at OER potential the Ni surface is likely oxidized and transformed to Ni oxide or oxyhydroxide. The question then arises on the relevance of the reported theoretical model, based on pure Ni.

Response to Comment-2a. We agree with the reviewer that the Ni surface would be oxidized at OER potential. Accordingly, we did careful MD simulations and found that the surface oxidation would not affect our conclusions regarding effects of O₂ on OER. The details are as follows:

It is difficult to determine the oxidation degree at Ni surface. We tried two limiting cases: pristine Ni and fully surface oxidized Ni (termed as NiO-Ni). Based on our original MD model of pristine Ni, we set up a MD model for the second case:

New Supplementary Figure 16. a. MD model for NiO-Ni surface. **b.** A representative unit cell structure of NiO-Ni surface (rotated by 90°).

After equilibrium, the ions distribution next to the NiO-Ni surface under oxygen-presence condition is presented below:

New Supplementary Figure 17. Ion distribution in the solution for NiO-Ni surface case. a. the distribution of oxygen, OH^- and K^+ distribution. **b.** the distribution of oxygen distribution. **c.** the distribution of oxygen and OH^- distribution.

New Supplementary Figure 18. MD results. a. Relative number density of atoms as a function of distance from NiO-Ni surface under oxygen-presence condition. **b.** Relative number density of atoms as a function of distance from NiO-Ni surface under oxygen-absence condition. **c.** A comparison of OH^- relative number density distribution between the oxygen-presence and oxygen-absence conditions.

These results show that for the NiO-Ni system, the oxygen molecules still tend to distribute near the surface of NiO-Ni (Fig. S16 a), and reduce the density of absorbed OH⁻ (Fig. S16 c), which is consistent to the pristine Ni surface case. We have added these new data in Fig. S14, S15 and S16, and cited them in the revised main text.

In page 8, we added “Considering Ni surface would be oxidized during the OER process, MD simulations were performed to examine whether the surface oxidation would affect our conclusions. As it is difficult to determine oxidation degree at the Ni surface, we choose the fully oxidized case (NiO-Ni). The MD model and the unit cell structure of NiO-Ni are presented in Fig. S16. The ions distribution next to the NiO-Ni surface under oxygen-presence condition is presented in Fig. S17. Again, the oxygen molecules exhibit a strong absorption near the surface, leading to a reduction of OH⁻ density (Fig. S18).”

Comment-2b. It is also not clear which is the driving force for O₂ adsorption at the surface under these conditions, which is the key point of the paper.

Response to Comment-2b. The driving force for O₂ adsorption at the surface is the van der Waals (vdW) interaction. Details have been put in the original support information. For the referee’s convenience, we select some relevant information and put it in following:

“The vdW interactions among Ni, water, KOH, and oxygen molecules (O₂) were described by using Lennard-Jones (LJ) pairwise potential. The potential function has the form of:

$$E(r_{ij}) = 4\epsilon_{ij} \left[\left(\frac{\sigma_{ij}}{r_{ij}} \right)^{12} - \left(\frac{\sigma_{ij}}{r_{ij}} \right)^6 \right]$$

where, r_{ij} denotes the distance between atoms, ϵ characterizes the strength of the interaction and σ determines the distance at which the two atoms interaction are zero. The substitution index i and j denote the component atoms and ions.”

Based on Table 1 in the original support information, the parameters ϵ and the σ are summarized below for the interaction between O₂ molecule and Ni atom (O(O₂)-Ni), OH⁻ and Ni (O(OH⁻)-Ni, H(OH⁻)-Ni), O₂ and NiO (O(O₂)-Ni(NiO), O(O₂)-O(NiO)), OH⁻ and NiO (O(OH⁻)-Ni(NiO), H(OH⁻)-Ni(NiO), O(OH⁻)-O(NiO), H(OH⁻)-O(NiO))

New Supplementary Table 3. Parameters of the Lennard-Jones force fields for the interaction between several specific atoms using mixing rule.

	ϵ	σ
O(O ₂) - Ni	0.052	2.66
O(OH ⁻) - Ni	0.059	2.72
H(OH ⁻)-Ni	0	0
O(O ₂) - Ni(NiO)	0.052	2.66
O(O ₂) - O(NiO)	0.006	3.29
O(OH ⁻) - Ni(NiO)	0.059	2.72
H(OH ⁻)-Ni(NiO)	0	0
O(OH ⁻) - O(NiO)	0.007	3.35
H(OH ⁻)-O(NiO)	0	0

In the above table, the parameters σ and ϵ for O(O₂) interaction with Ni are very close to those for O(OH⁻) interaction with Ni. It suggests that O₂ and OH⁻ could occupy similar positions next to the Ni/NiO surface, reducing the amount of OH⁻ absorbed on the surface. This indeed is what we observed in our simulations.

To clarify the reason for O₂ absorbing near the surface of Ni/NiO, we have added a short discussion about the driving force for it and included the above Table in the support information.

In page 8, we added “The detrimental effect of oxygen should stem from the strong van der Waals (vdW) force for O(O₂) interaction with Ni/NiO, which is nearly the same to the interaction between O(OH⁻) and Ni/NiO (Supplementary Table 3). In our MD simulations, the O₂ and OH⁻ occupy similar positions next to the Ni/NiO surface. This reduces the amount of OH⁻ in the EDL.”

Comment-3. Also assuming that the model is realistic, it is not clear what the rationale of varying the substrate charge is. Which is the point that the authors want to make? The authors should also check the reported charges, as in the text and in the caption they are different in sign.

Response to Comment-3. We thank the reviewer for this critical comment. To address this issue, in page 7, we added “During the OER process, the Ni surface should have positive charges. The Coulomb force could drive OH⁻ closer to Ni surface, eliminating the detrimental role of O₂. To examine this effect, a set of surface charges was induced in the system (Supplementary method S7, Fig. S15).” The values of charges induced are based on the surface capacitance value of Ni. Details have been put in the original support information. For the referee’s convenience, we selected some relevant information and put it in following:

“In our simulations, different charging conditions of the active materials (Ni) were investigated. Given that the surface capacitance value of Ni is 20 $\mu\text{F}/\text{cm}^2$ and electric potential window is chosen as 1 V, the maximum charge per target Ni atom (upper Ni atoms

which are exposed to electrolyte) is calculated to be 0.033e. Accordingly, 0, +0.0083, +0.0167, +0.025 and +0.033e were imposed on each target Ni atom respectively to consider different charging conditions.”

Comment-4. The English should be improved prior to publication.

Response to Comment-4. We thank the reviewer very much for the valuable suggestion. We have modified our manuscript carefully to improve.

Response to Reviewer-2

We thank the reviewer for the thoughtful and encouraging comments about our manuscript, and welcome the opportunity to address and clarify the issues raised in the report. Our responses to points raised in the report are described below following specific reviewer comments.

General Comment. This is a very interesting paper of very high scientific quality, topicality and general interest. The measurement of oxygen evolution at a single nano sheet is very nice and the experimental data are of very high quality. This in itself adds to the quality of the paper.

Response to the General Comment. We thank the reviewer for the very positive summary of our work and for recognizing the high quality of experimental data.

Comment-1. The authors have noticed that the Tafel slope is reduced significantly if oxygen is excluded from the solution. They suggest that adsorbed oxygen substitutes for adsorbed hydroxyl ion and thus reduced the catalytic activity. This may well be so. However, It has been commonly observed in linear potential sweep measurements performed in aqueous solution in the active oxygen evolution region, that the Tafel slope increases as the over potential is increased. This increase has been attributed to gas bubble shielding effects which add a resistance to the interface thereby changing the Tafel slope. Indeed one can increase the potential window over which a low Tafel slope is observed by devising a route to dislodge and expel gas bubbles from the electrode surface. Consequently, I am not clear as to how the authors excluded the oxygen. Was it by simple degassing? The authors should elaborate here.

Response to Comment-1. We appreciate the constructive comments raised by reviewer towards the oxygen effect on the Tafel slope change. In this work, we expelled the oxygen bubbles by simple degassing. Before illustrating the detail about it, we want to present the reason why we use degassing method. We agree with reviewer's opinion that dislodging and expelling gas bubbles from the electrode surface can enhance the catalyst performance. However, it is a technique difficulty for us to expel gas bubbles continuously along with the OER process in a nanodevice. We tried to use microfluidics technique to expel oxygen bubbles, but the motive power to drive fluidics flow continuously in the nano-device is a challenge for us. Thus, in this work, we firstly prepared electrolyte absent with oxygen and then drop it on top of the nanodevice. As only a small quantity of electrolyte is needed for the nano-device, a simple degassing method was used to exclude the oxygen in this work.

We have added this process in the new revised manuscript.

In page 10, "Two different electrolytes were adopted with different contents of O₂, namely 0.1 M KOH saturated with O₂ and 0.1 M KOH absent of O₂. Specifically, the electrolyte absent with O₂ was prepared using degassing method where N₂ was inserted into the 5mL 0.1M KOH electrolyte for 1 hour to expel the O₂."

Comment-2. Furthermore, if the catalysis occurs primarily at edge sites rather than basal plane sites, the reaction rate should depend on flake size. Has this effect been noted? I am happy that the paper get accepted if these queries are addressed.

Response to Comment-2. We thank the reviewer's kind concern about edge effect on the catalyst performance. Indeed, for some catalysts, its catalysis performance is significantly affected by the number of catalytically active edge sites, for example, MoS₂.^{R1, R2} For the edge effect in this work, we discuss from two aspects in the following:

First, if the reviewer referred to the graphene edge, we believe the edge effect should be minor. It is known that graphene nanosheet itself has negligible OER performance compared with metallic catalysts.^{R3, R4, R5} In addition, the graphene used in our device has a relatively large flake size, > 1 micron. The edge should be minor in comparison with basal plane.

Second, if the reviewer referred to the edge effects of Ni nanoparticles, the edge effect is also minor, because it has been reported that for Ni particle, there is no obvious edge effect.^{R6}

So, we conclude a minor edge effect in our experiments. We also want to point out that since the same device was used to investigate the effects of oxygen, the edge effect will not change our conclusions.

Reference:

R1. Xie, J. et al. Defect - rich MoS₂ ultrathin nanosheets with additional active edge sites for enhanced electrocatalytic hydrogen evolution. *Advanced Materials* **25**, 5807-5813 (2013).

R2. Kibsgaard, J., Jaramillo, T. F. & Besenbacher, F. Building an appropriate active-site motif into a hydrogen-evolution catalyst with thiomolybdate [Mo₃S₁₃]²⁻ clusters. *Nature Chemistry* **6**, 248-253 (2014).

R3. Vasileff, A., Chen, S. & Qiao, S. Z. Three dimensional nitrogen-doped graphene hydrogels with in situ deposited cobalt phosphate nanoclusters for efficient oxygen evolution in a neutral electrolyte. *Nanoscale Horizons* **1**, 41-44 (2016).

R4. Faisal, S. N. et al. Pyridinic and graphitic nitrogen-rich graphene for high-performance supercapacitors and metal-free bifunctional electrocatalysts for ORR and OER. *RSC Advances* **7**, 17950-17958 (2017).

R5. Liang, Y. et al. Co₃O₄ nanocrystals on graphene as a synergistic catalyst for oxygen reduction reaction. *Nature Materials* **10**, 780-786 (2011).

R6. Nørskov, J. K. et al. The nature of the active site in heterogeneous metal catalysis. *Chemical Society Reviews* **37**, 2163-2171 (2008).

Response to Reviewer-3

We thank the reviewer for the thoughtful and encouraging comments about our manuscript, and welcome the opportunity to address and clarify the issues raised in the report. Our responses to the points raised in the report are described below following specific reviewer comments.

General Comments. The authors in this work propose to use on-chip electrochemical, EIS and I-V measurement to monitor the catalytic interface of Ni nanoparticle decorated graphene during OER reactions. The authors utilize a single sheet nanodevice with three working electrodes and necessary electrochemical gating setup to achieve the measurements. As a result, on-chip OER reactions, corresponding impedance and conductance measurements can be carried out in two different electrolytes, with or without dissolved oxygen. Characterization of electrochemical interface during practically relevant electro catalytic reactions such as HER, OER, and ORR are indeed challenging, especially for in situ and in operando studies. On-chip based concurrent electrochemical and electrical characterization could be an effective strategy. Therefore, the topic of this work is of great novelty and significance, and fits the broad interests of Nat Comm readers.

Here are some of my concerns, comments, and suggestions, which should be addressed before this manuscript can be considered acceptable for publication.

Response to the General Comment. We thank the reviewer for very positive summary of our work and for recognizing the substantial advance of the nano-device to study the electrochemical interface. We also appreciate the need to expand our citations of previous work clarifying the current signal interfere during the testing, and the detailed changes made in the revised manuscript below.

Comment-1. Although it has been demonstrated previously in literature that measurements based on nanodevice can be helpful for the studies of electrocatalytic systems, the authors in this manuscript, however, did not offer a convincing argument that the experimental design and technique used in this work is better than approaches otherwise available. For example, the OER and EIS measurements can be easily done with the same material on a bulk electrode, which probably will offer the same conclusion revealing the effect of O₂ concentration on the catalyst performance and electrochemical interface. The I-V measurement is only available based on the on-chip platform. However, the authors only declared that the result is in consistent with EIS. To clearly demonstrate the scientific significance of their approach, the authors are suggested to offer discussions on what different information the in-operando I-V or I-Vg measurements can reveal compared with EIS measurement, and on the analysis and interpretation of the data.

Response to the comment-1. We thank the reviewer for this critical comment. The following is a discussion regarding key differences of I-V measurement in our nano-device compared with the normal EIS measurements.

First, EIS measurement may not have sufficient resolution to detect the resistance change of electrode materials during OER process. EIS is an impedance measurement. It is known that the EIS signal measured at high frequency (in theory, infinitely large) represents the resistance. However, in our experiment, the signals at the highest frequency are nearly identical, which provide no useful information. In contrast, the I-V measurement can directly test the resistance change of the electrode materials under different conditions.

Second, the EIS measurement cannot achieve an in-situ monitoring as it is time-consuming compared with I-V measurement.

Third, the EIS measurement may provide inaccurate information. As the EIS measurement cannot directly provide the value of resistance, an equivalent circuit model should be used to interpret EIS data. However, it is well known that the equivalent circuit model is not unique. There could be several equivalent circuit models that can fit the EIS data equally well. This would lead to uncertainties in the analysis. Our I-V measurements do not suffer this problem.

We have revised the main text of the manuscript and added a short discussion on it.

In page 6, “Our I-V measurement approach has several advantages over EIS. First, our EIS data show nearly no differences at the highest frequency under the oxygen-presence and oxygen-absence conditions. It could not provide sufficient resolution to make a comparison of the electrode resistance under these two conditions. Second, EIS measurement is time-consuming, which is not feasible for in-situ measurement. Third, a relative complicated equivalent circuit model is required to interpret EIS results.”

Comment-2. The significance of the major conclusion of this manuscript is not clearly discussed. The conclusion that O_2 level can affect the adsorption of OH and hence the overall performance of the OER catalysts is in accordance with the Le Chatelier's principle that as the end product, lower concentration of O_2 will indeed facilitate the OH to O_2 transition.

Response to the comment-2. We thank reviewer for constructive advice about the Le Chatelier's principle applied in the OER process. We revise our manuscript by adding in page 2:

“Oxygen is a product of OER. In terms of reaction equilibrium (i.e., Le Chatelier's principle), it is known that O_2 concentration increase in electrolyte (with the ongoing of OER) hinders the catalytic reaction. But there is no clear answer whether oxygen molecules would affect OER kinetics.”

Comment-3. The electrical measurement should be precise and accurate even at the nA level. But the electrical data in Fig. 3 is very noisy even when compared with the similar works previously published by the same author (on the topic of single nanowire battery).

Response to the comment-3. We thank the reviewer for raising this point. In next Figure, we compare the results in this paper with those that the reviewer referred.

Comparison of I-V curves in different works. **a, b.** I-V curves of different nanowire part before and after the electrochemical process respectively in Xuxu's work. **c, d.** I-V curves of Single vanadium oxide nanowire after Li^+ ion intercalation/ iondeintercalation respectively in Liqiang Mai's work. **e, f.** I-V curves in this work.

The I-V measurement in this work was performed in-situ along with the OER dynamic process, while our two other previous works were tested under static state. In addition, the applied voltage range in this work, 4mV, is much smaller than previous work, 2000 mV and 8 V.^{R1, R2} All these could result in apparently noisy I-V curves in this work. We should point out that the I-V result differences between the oxygen-absence and oxygen-presence

conditions are sufficiently larger than accuracy of measuring facility. We can draw reliable conclusions.

Comment-4. For the temporary (in situ) I-V measurement, could there be some interference of the current signals, since WE1 is used to carry out OER reaction (electrochemical current), and WE2 and WE3 is used to carry out conductive current, and the three electrodes are electrically connected by the nanosheet. If we view the system as an electrochemically gated transistor, then the OER current can be viewed as gate leakage current. Such current, concurrently measured with I-V current between WE2 and WE3, should be demonstrated. Unless the gate current through WE1 is several orders smaller than the conductive current through WE2 and WE3, then one need to take into account the current interference. Background subtraction may be necessary to reveal the true conductive current level.

This is also the reason why in situ electrical measurement is technically challenging, because when the reaction is in place, the concurrent electrical measurement is easily being interfered. Note that in Manish Chhowalla's work (Nat Mater, 2015, 15, 1003), similar experimental set-up was used to study the MoS₂ HER reactions, but all the I-V measurement that determines the contact resistance is done ex situ, considering the possible interference as well. This paper should also be an appropriate citation in line 119 of this manuscript as well. For another example, in Xiangfeng Duan's work (ACS Nano, 2016, 10, 9919), the interference of similar level electrochemical gating current on the concurrent conductive current is considered, and an equivalent circuit is used to help subtract the background.

Response to the comment-4. We thank reviewer for this critical concern about the interferences of current signals. We carefully studied the Xiangfeng Duan's works.^{R3,R4} Constructed by his work, the equivalent current circuit in our work was carried out based on our testing method. For the referee's convenience, we selected some relevant figures in Xiangfeng Duan's work and put it together with our equivalent current circuit of this work:

On-chip electrical transport spectroscopy (ETS) comparison. **a.** schematic illustration of the concurrent measurements of electrochemical (I_G) and electrical transport (I_{sd}) characteristics of MR-1 in Xiangfeng Duan’s work. **b.** Equivalent circuit model of the ETS (gate) measurement in Xiangfeng Duan’s work. **c.** schematic illustration of the concurrent measurements of electrochemical (I_G) and electrical transport (I_{sd}) characteristics of this work. **d.** Equivalent circuit model of the ETS (gate) measurement in this work.

The measurement setup of this work has some differences from that in Xiangfeng Duan’s work. First, in this work, there are three separated electrodes (as source, drain and working electrode respectively) on the substrate. In Xiangfeng Duan’s work, there are only two electrodes, and thus there is a sharing electrode for I-V measurement and electrochemical measurement. Second, there are two independent current circuits in this work. An electrochemical workstation (Autolab PGSTAT 302N) and a semiconductor parameter analyzer (Agilent, B1500A) were used separately to monitor the signals of the current of OER process and conductive current of I-V process. Therefore, we believe there is no necessary to subtract the background in this work.

We added the equivalent current circuit in support information and cited the Xiangfeng Duan’s works (Ref. 40, revised manuscript) and Manish Chhowalla’s work (Ref. 29, revised manuscript).

In page 10, we added “Figure S19 shows the schematic illustration of the concurrent measurement of this work, and the equivalent circuit model of the electrical transport spectroscopy (gate) measurement.”

Comment-5. In Fig. 3b and c, it looks like the I-V for O₂ and no O₂ are different at 0V while overlapping at 0.7V. Does this indicate that the difference of OH adsorption is different under no OER reactions (0V), while OH adsorption is the same when OER reaction occurs (0.7V)? By the way, the signs of voltage axis seem to be wrong.

Response to the comment-5. The reviewer's understanding is right. The influence of O₂ on the OH⁻ ions adsorption is different under different OER potentials. This is due to the competition between van der Waals (vdW) force and Coulomb force. Under small OER potential close to 0 V, the vdW force plays a dominant role. Since the vdW interaction between O₂ and Ni/NiO surface is almost the same to that between OH⁻ ion and Ni/NiO surface (New Supplementary Table S3). O₂ and OH⁻ would occupy similar positions next to the Ni/NiO surface, reducing the amount of adsorbed OH⁻ ions. This indeed is what we observed in simulations (Fig. 4a, b of original manuscript). For reviewer's convenience, we show the figure below:

New Supplementary Table 3. Parameters of the Lennard-Jones force fields for the interaction between several specific atoms using mixing rule.

	ϵ	σ
O(O ₂) - Ni	0.052	2.66
O(OH ⁻) - Ni	0.059	2.72
H(OH ⁻)-Ni	0	0
O(O ₂) - Ni(NiO)	0.052	2.66
O(O ₂) - O(NiO)	0.006	3.29
O(OH ⁻) - Ni(NiO)	0.059	2.72
H(OH ⁻)-Ni(NiO)	0	0
O(OH ⁻) - O(NiO)	0.007	3.35
H(OH ⁻)-O(NiO)	0	0

Data in original manuscript. a. Relative number density ρ of different electrolyte ions as a function of distance from the Ni cathode surface with oxygen concentration of 0. **b.** Relative number density ρ of different electrolyte ions as a function of distance from the Ni cathode surface with oxygen concentration of 0.12 mmol/cm³.

However, at higher OER potential, the Ni/NiO surfaces should have positive charges. The enhanced Coulomb force would drive OH⁻ closer to the charged Ni/NiO surfaces, eliminating the detrimental role of O₂. To examine this effect, a set of surface charges was induced in our simulations. In Fig. 4e (shown below), there is no difference of OH⁻ adsorption when the surface charge is close to 0.03e/Ni-atom.

Data in original manuscript. The relative density of OH⁻ ions in the stern layer as a function of the charge number for per Ni atom.

In the end, we would thank reviewer for pointing out the problem of voltage signs. We have updated the figure in revised manuscript.

Comment-6. Based on Equation (1), the physical model of this device is a graphene based transistor. Which physical parameter in equation (1) is directly linked to the measured current change, and how? Based on the description in the manuscript, the authors attributed the current change to the change of carrier densities as well as change in double layer capacitance. Which one is more dominant? And if both are in the case, can these signals be conclusively linked to the event of OH adsorption on Ni nanoparticles? I feel that more detailed discussions on the origin of the signals, the interpretation of the data, and how the data is linked to the conclusions need to be provided.

Response to the comment-6. We are sorry for the confusion in our previous manuscript. To clearly explain our model, we show the equivalent electric circuit model in next figure.

a. Schematic diagram of electrochemical gate tuned nano-device. **b.** SEM image of the as-prepared Ni-Graphene nanosheets. **c.** TEM image of the as-prepared Ni-Graphene nanosheets.

As stated in our manuscript, we attribute the current change to the gating effect of graphene. The left side of Eq. (1) is the voltage applied on the whole device including graphene and the Ni nanoparticles.^{R5} As shown in above figure, this total voltage is applied on two serially connected capacitors. The first capacitor is from the EDL, i.e., adsorption of OH⁻ at Ni or graphene surface. The second capacitor is the gate voltage applied on graphene. The right side of Eq. (1) is summation of the two voltages applied on these two capacitors. Relative magnitude of these two voltages depends on the capacitance values. When the EDL capacitance C_{dl} is reduced owing to the adsorbed oxygen molecules (this will be addressed in detail later), the voltage applied on EDL is increased and thus the gating voltage on graphene is reduced. As is well known, the carrier density of graphene (n in Eq. (1)) is lower at a lower gating voltage.^{R5, R6} Thus, the resistance of graphene will be increased, as we seen in experiments (Fig. 3C).

In following, we address the specific comments.

- (1) Carrier density n in Eq. (1) is the quantity directly related to the measured current in experiments.
- (2) Capacitance change and carrier density are cause and consequence.
- (3) The change of capacitance C_{dl} is directly related to the OH⁻ and oxygen adsorption on Ni surfaces.

The reduction of capacitance C_{dl} under the oxygen-presence condition has been comprehensively studied in this work by using molecular dynamics simulations. We have

studied EDL at three different surfaces, Ni (in previous manuscript), NiO, and graphene. As seen in Figure 4 and two following figures, the oxygen molecules occupy similar positions of OH⁻ ions and reduce the adsorption of OH⁻ ions. As a result, the EDL capacitance will be reduced.

New Supplementary Figure 15. a. Adsorption of OH⁻ ion next to NiO-Ni surface under oxygen-presence condition. **b.** A comparison of OH⁻ adsorption under oxygen-presence and oxygen-absence condition.

A comparison of OH⁻ adsorption next to graphene surface under oxygen-presence and oxygen-absence condition.

In page 6, we have revised the main text to have a detailed discussion on the results of I-V measurement. “The physical principle behind our I-V measurement is the gating effect of graphene. The electrochemical potential applied on our device can be decomposed into two parts (Figure S11). One is applied on the electric double layer at the electrode-electrolyte interface. The second one is the voltage applied on graphene.”;

“The first term on the right of Eq. (1) describe the voltage applied on graphene and its relationship to carrier density n . The second term describe the relationship between EDL voltage and EDL capacitance.”

“We hypothesize that oxygen would lead to a reduced double layer capacitance (C_{dl}). As a result, the EDL voltage (second term in Eq. (1)) will increase. Provided a constant value of V_G in our experiments, the carrier density n in the electrode would drop (first term in Eq. (1)) and the electrode resistance increases. This could explain the observed electrode resistance changes in our I-V experiments (Fig. 3b).”

Comment-7. The overall writing of the manuscript can be improved. Phrase like “...are rarely in-depth understood”, “in traditional, ...” and some other sentences can be further polished for an improved quality. There are also minor errors in supporting materials that interfere with smooth reading and understanding of the materials. A few examples include: in Figure S1e, “Left off” should be “lift off”?; in caption of Figure S4, “rietveld” may be a typo; in figure s6- it is better to note which color represents which scan rate in Figure S6a and Figure S6b; in Figure S13 caption, “with oxygen in the electrolyte (left)” should be “without oxygen in the electrolyte (left)”?

Response to the comment-7. We are sorry for these typos and mistakes. We have revised these in the new support materials.

The new Figure S1:

The Figure S4 and its new caption:

Figure S4. A RIETVELD refinement of XRD patterns. a. Ni particles; b. graphene.

The new caption for the Figure S13:

“Figure S13. a, b, c, and d. Relative number density ρ of different electrolyte ions as a function of distance from the Ni cathode surface with a charge of 0, 0.0250, 0.0167, 0.033 e/Ni-atom without oxygen in electrolyte (left) compared with the oxygen concentration of 0.12 mmol/cm³ (right).”

References:

- R1. Xu, X. *et al.* In situ investigation of Li and Na ion transport with single nanowire electrochemical devices. *Nano Letters* **15**, 3879-3884 (2015).
- R2. Mai, L., Dong, Y., Xu, L. & Han, C. Single nanowire electrochemical devices. *Nano Letters* **10**, 4273-4278 (2010).
- R3. Ding, M. *et al.* Nanoelectronic Investigation Reveals the Electrochemical Basis of Electrical Conductivity in *Shewanella* and *Geobacter*. *ACS Nano* **10**, 9919-9926 (2016).
- R4. Ding, M. *et al.* An on-chip electrical transport spectroscopy approach for in situ monitoring electrochemical interfaces. *Nature Communications* **6** (2015).
- R5. Das, A. *et al.* Monitoring dopants by Raman scattering in an electrochemically top-gated graphene transistor. *Nature Nanotechnology* **3**, 210-215 (2008).
- R6. Novoselov, K. S. *et al.* Electric field effect in atomically thin carbon films. *Science* **306**, 666-669 (2004).

REVIEWERS' COMMENTS:

Reviewer #1 (Remarks to the Author):

The authors have addressed the concerns that I had on their original submission and added additional experimental data to confirm and validate their findings. I found the revised version more convincing.

I still believe that the novelty of the paper is on the reported technique, more than the results on the effect oxygen on the OER activity.

On this latter, the discussion on thermodynamics vs kinetics of the reaction (and their relation with Le Chatelier principle) could be made more clear and rigorous.

Also, the revisions made on the manuscript (i.e. when adding some comments as suggested by the reviewers) should be better integrated in the text.

Reviewer #2 (Remarks to the Author):

I have read the revised manuscript and the response to my original comments and concerns. These have all been addressed by the authors. I am happy that the paper be published.

Reviewer #3 (Remarks to the Author):

The authors have carefully and clearly answered all my questions, and strengthened their claims by providing extra experiments, discussions, and clarifications. At this point I have no additional comments on the content. I believe there is indeed substantial advance of the nano-device approach to study the electrochemical interface, and this work should be considered publishable in Nat Commun.

Response to Reviewer-1

General Comment. The authors have addressed the concerns that I had on their original submission and added additional experimental data to confirm and validate their findings. I found the revised version more convincing.

Response to the General Comment. We thank the reviewer for the helpful comments and remarks and feel that the manuscript benefitted greatly.

Comment-1. I still believe that the novelty of the paper is on the reported technique, more than the results on the effect oxygen on the OER activity.

On this latter, the discussion on thermodynamics vs kinetics of the reaction (and their relation with Le Chatelier principle) could be made more clear and rigorous.

Response to Comment-1. We thank the reviewer for the insightful suggestion. Indeed the novelty of our designed technique is one critical point of this paper. Note that the advantages of this technique have been explained in section ‘Resistance testing and analysis’.

The part about the thermodynamics and kinetics of the reaction has been revised in the main text. In page 2, we added ‘However, there are much fewer studies on OER kinetic process, particularly those processes taking place in the electrode/electrolyte interface region. The distribution of ions and water at the interface determine the kinetics of mass and electron transfer process. Our understanding is still quite limited. Oxygen is a product of OER. In terms of reaction equilibrium (i.e., Le Chatelier’s principle), it is known that O₂ concentration increase in electrolyte (with the ongoing of OER) hinders the catalytic reaction. But there is no clear answer whether oxygen molecules would affect OER kinetics.’

In discussion part (page 9), we added ‘where Ni is not able to immediately oxidize water. Ni transforms into a highly reactive species by reacting with OH⁻ and then generates O₂. From thermodynamic aspect, the increased O₂ concentration in reaction system leads to a higher Gibbs free energy, thereby increasing the onset potential under oxygen-presence condition. It could also be understood in terms of Le Chatelier’s principle. However, the reaction kinetic processes are not considered in above two thermodynamic aspects. The kinetic process includes the transport of the reactants to an active site, the adsorption of the reactions to the active site, and the reaction of reactants to form an adsorbed product. Using EIS, temporal I-V measurement results, and MD simulation, we can conclude that O₂ can adsorb near the surface of catalyst driven by the strong vdW interaction. Their position overlaps with that of adsorbed OH⁻ ions, reducing the concentration of OH⁻ in EDL, thus slowing down the charge transfer process and OER reaction kinetics. Different from the Le Chatelier’s principle, which provides a clue to understand reaction direction from thermodynamic perspective, our new understanding offers insights on how surface adsorbed O₂ could influence the reaction kinetics.’

Comment-2. Also, the revisions made on the manuscript (i.e. when adding some comments as suggested by the reviewers) should be better integrated in the text.

Response to Comment-2. We thank the reviewer for the constructive suggestion. We have carefully revised our manuscript to make it more clear and fluent. Some essential modifications are shown below.

1. To make the aim of carrying out additional conventional setup measurement more clear, in page 5, we added ‘To check whether the results obtained from this nanodevice is consistent with conventional powder sample performance, Supplementary Fig.7 a-c compare OER performance of Ni catalysts that were prepared in conventional ways.’

2. To make presentation of MD simulation part more logic, in page 8, we changed the orders of these two paragraphs. ‘Considering Ni surface would be oxidized during the OER process, other MD simulations were performed to examine whether the surface oxidation would affect our observations based on the above pure Ni theoretical model. ...’ and ‘During the OER process, the Ni surface should have positive charges. The Coulomb force could drive OH⁻ closer to Ni surface, eliminating the detrimental role of O₂. ...’

3. In page 5, we revised ‘There are clear performance gaps under oxygen-absence or oxygen-presence condition. They qualitatively agree with the results from our nanodevice.’

4. We also revised in page 8, ‘Compared with Fig. 4a, Fig. 4b clearly demonstrates that O₂ adsorbed at interface, where their position overlaps with that of the adsorbed OH⁻, and significantly hindered accumulation of OH⁻ in EDL. The increase of O₂ concentration leads to a monotonic decrease of OH⁻ density at the reaction interface (Fig. 4d, Supplementary Fig. 15), leading to the decrease of C_{dl}’

‘The detrimental effect of oxygen should stem from the strong van der Waals (vdW) force between O(O₂) and Ni/NiO, which is nearly the same to the interaction between O(OH⁻) and Ni/NiO. (Supplementary Table 3). The similar interaction force drives the O₂ and OH⁻ occupy similar positions next to the Ni/NiO surface. This reduces the amount of OH⁻ in the EDL.’

Response to Reviewer-2

General comment. I have read the revised manuscript and the response to my original comments and concerns. These have all been addressed by the authors. I am happy that the paper be published.

Response to the General Comment. We thank the reviewer for the supportive comments of our manuscript.

Response to Reviewer-3

General comment. The authors have carefully and clearly answered all my questions, and strengthened their claims by providing extra experiments, discussions, and clarifications. At this point I have no additional comments on the content. I believe there is indeed substantial advance of the nano-device approach to study the electrochemical interface, and this work should be considered publishable in Nat Commun.

Response to the General Comment. We thank the reviewer for the productive discussion and positive comment.